# Regulation of DNA Replication Licensing and Re-Replication by Cdt1

**DOI:** 10.3390/ijms22105195

**Published:** 2021-05-14

**Authors:** Hui Zhang

**Affiliations:** Department of Chemistry and Biochemistry, Nevada Institute of Personalized Medicine, University of Nevada, Las Vegas, 4505 South Maryland Parkway, Box 454003, Las Vegas, NV 89154, USA; hui.zhang@unlv.edu; Tel.: +1-702-774-1489; Fax: +1-702-895-4072

**Keywords:** DNA replication, DNA repair synthesis, Cdt1, Replication licensing, Cdt2, PCNA, CRL4^Cdt2^, DNA re-replication, genome instability

## Abstract

In eukaryotic cells, DNA replication licensing is precisely regulated to ensure that the initiation of genomic DNA replication in S phase occurs once and only once for each mitotic cell division. A key regulatory mechanism by which DNA re-replication is suppressed is the S phase-dependent proteolysis of Cdt1, an essential replication protein for licensing DNA replication origins by loading the Mcm2-7 replication helicase for DNA duplication in S phase. Cdt1 degradation is mediated by CRL4^Cdt2^ ubiquitin E3 ligase, which further requires Cdt1 binding to proliferating cell nuclear antigen (PCNA) through a PIP box domain in Cdt1 during DNA synthesis. Recent studies found that Cdt2, the specific subunit of CRL4^Cdt2^ ubiquitin E3 ligase that targets Cdt1 for degradation, also contains an evolutionarily conserved PIP box-like domain that mediates the interaction with PCNA. These findings suggest that the initiation and elongation of DNA replication or DNA damage-induced repair synthesis provide a novel mechanism by which Cdt1 and CRL4^Cdt2^ are both recruited onto the trimeric PCNA clamp encircling the replicating DNA strands to promote the interaction between Cdt1 and CRL4^Cdt2^. The proximity of PCNA-bound Cdt1 to CRL4^Cdt2^ facilitates the destruction of Cdt1 in response to DNA damage or after DNA replication initiation to prevent DNA re-replication in the cell cycle. CRL4^Cdt2^ ubiquitin E3 ligase may also regulate the degradation of other PIP box-containing proteins, such as CDK inhibitor p21 and histone methylase Set8, to regulate DNA replication licensing, cell cycle progression, DNA repair, and genome stability by directly interacting with PCNA during DNA replication and repair synthesis.

## 1. Introduction

DNA replication licensing is tightly controlled to prevent re-replication of the genome during cell cycle progression in eukaryotic cells [1,2,3]. In metazoans, a major regulatory mechanism by which only one round of DNA replication occurs in S phase is the control of the activity or protein levels of the replication licensing protein Cdt1 [1,4,5,6,7] in the cell cycle. Recently, progress revealed that Cdt1 and DNA replication licensing are regulated at multiple levels during cell cycle progression [7,8,9,10,11], providing new insights into the mechanism by which the fidelity of DNA replication and genome stability are controlled in the cell cycle.

In single cell eukaryotic organisms such as budding yeast *Saccharomyces cerevisiae*, DNA replication initiates at multiple autonomously replicating sequences (ARSs) in the S phase of a mitotic cell cycle [12]. These ARSs are specifically recognized by a replication initiator protein complex, the Origin Recognition Complex (ORC) consisting of six proteins (Orc1-Orc6), in an ATP-dependent binding process to initiate the bidirectional DNA replication [13]. In addition, Cdc6 and the Mcm2-7 helicase proteins are also found essential for the initiation of DNA replication in the budding yeast [3,14]. While the Orc1-6, Cdc6, and Mcm2-7 proteins are essential for DNA replication initiation and highly conserved from yeast to human [14], Cdt1 is initially identified in fission yeast *Schizosacharomyces pombe* [15], which is also required for DNA replication initiation [3]. The budding orthologue of Cdt1, encoded by *Tah11/Sid2*, is subsequently identified as the *tah11/sid2* mutants that fail to load Mcm2-7 to ARS to initiate DNA replication but not the DNA replication elongation, resulting in the loss of minichromosomes in an ARS-number-dependent manner [16].

## 2. The Assembly of the Pre-Replicative Complex (Pre-RC) for DNA Replication Licensing

In multi-cellular organisms, DNA replication initiation also occurs at multiple replication origins along the chromatin DNA in cells undergoing replication and mitotic cell divisions, although the number and exact locations of many DNA replication origins vary in different cells [17,18,19]. The firing of various DNA replication origins is further regulated in time and space during S phase progression, affected by chromatin structure, DNA and histone modifications, and transcriptional activities [17,18,20,21]. The early replication origin regions typically replicate the genomic region in the early S phase, whereas late replication regions usually synthesize their DNA in the late S phase [18]. However, all DNA replication origins are fired once and only once in a mitotic cell cycle to ensure that fully replicated DNA at the end of S phase is strictly followed by a G2 phase and a mitotic cell division to produce two daughter cells with the same genome DNA contents as the mother cell [1,3]. The competence for the next round of DNA replication, also called DNA replication licensing, is established in late mitosis and early G1 [3,22,23,24]. This replication licensing process requires the assembly of pre-RC on the ORC protein-bound replication origins, including the recruitment of Cdc6, Cdt1, and the Mcm2-7 helicase complex onto the replication origin-ORC complexes. However, the assembled Mcm2-7 complex in the pre-RC does not have the replication helicase activity at this stage. Subsequent activation of the Mcm2-7 helicase complex requires the activities of two kinases: the cyclin-dependent kinase (cyclin/CDKs) and Dbf4-dependent kinase (DDK) at the onset of the S phase, leading to the unwinding of DNA replication origins to initiate DNA replication in the S phase [1,3,17].

Analyses of step-wise in vitro biochemical assembly of replication initiation complexes using purified replication initiation proteins, and the establishment of yeast replication initiation process, reveal that the Orc1-5 and Cdc6 protein complex forms a ring-clamp structure to encircle the double-stranded replication origin DNA in an ATP-dependent process [3,25,26]. The tertiary ORC-Cdc6 complex next loads the pre-existing Cdt1-Mcm2-7 protein complex onto the DNA replication origin DNA to form the pre-RC [7]. In budding yeast, Orc6, the only ORC protein not required for DNA binding, interacts with Cdt1 to load Mcm2-7 helicase onto the origin-containing DNA [27]. The ORC complex lacking Orc6 fails to interact with Cdt1 and is defective to load the Mcm2-7 helicase, consistent with the observation that in vivo depletion of Orc6 inhibits pre-RC assembly and maintenance. CDK phosphorylation of Orc6 blocks Cdt1 binding and the Mcm2-7 helicase loading. Recent studies showed that replication licensing requires multiple Cdt1 proteins at each DNA replication origin to repeatedly load the single Mcm2-7 hexameric complex onto the origin-bound ORC-Cdc6 complex to eventually form the head-to-head double hexameric Mcm2-7 helicase complexes. The active double hexameric Mcm2-7 helicase complexes encircle the DNA double helix to unwind the DNA duplex [27,28,29]. The loading of Mcm2-7 onto the DNA requires the entry of the DNA into the ring-like Mcm2-7 clamp and this process is gated by a gap between Mcm2 and Mcm5 [28]. Cdt1 is required for the loading of the hexameric Mcm2-7 complex because it can overcome the block conferred by an auto-inhibitory domain in Mcm6 that prevents the recruitment of Mcm2-7 to the ORC-Cdc6 complex [27]. In addition, Cdt1 interacts with Mcm2, 6, and 7 to destabilize the interface between Mcm2 and Mcm5 to maintain the Mcm2-7 in an open ring structure for Mcm2-7 helicase loading onto the duplex DNA [28]. The Cdt1-Mcm2-7 complex further activates the Orc1-Cdc6 ATPase activities to hydrolyze ATP to promote Mcm2-7 helicase loading [30]. The Cdc6 ATPase activity is also required to release Cdt1 to form the ORC-Cdc6-Mcm2-7 complex, which functions to serve as the platform for the assembly of the Mcm2-7 double hexameric complex [31]. The ORC-Cdc6-Mcm2-7 complex is negatively regulated by cyclin/CDK-mediated phosphorylation of the ORC proteins, such as Orc2 and Orc6, to prevent premature activation of Mcm2-7 helicase [27]. The activation of the helicase activity of the Mcm2-7 complex also requires the association with Cdc45 and Gins proteins, as well as Mcm10, to form the Cdc45-Mcm2-7-Gins (CMG) complex that unwinds the duplex DNA during DNA replication [3,32,33].

In yeast, the ORC complex is essential for marking the specific replication origins (the ARS sequences) and loading the Cdt1-Mcm2-7 complex onto the origins for DNA unwinding at the onset of DNA replication [13]. Whether the ORC complex and Cdc6 can assemble the functional pre-RCs without the specific origin DNA sequences has been tested by expressing and tethering GAL4-fusion proteins with individual Orc protein or Cdc6 to GAL4 binding sites in human cells [34]. These tethering experiments showed that the recruitment of GAL4-Orc or Cdc6 fusion proteins to the GAL4 binding sites is sufficient to create artificial replication origins for DNA replication. These studies indicate that binding of the ORC complex or Cdc6 protein to DNA is sufficient to recruit the Cdt1-Mcm2-7 helicase complex for replication licensing.

However, recent studies revealed that not all conserved ORC proteins are essential in human cells. Biallelic deletions of two human ATPase ORC protein encoding genes, Orc1 or Orc2, indicate that human Orc1 or Orc2 are dispensable for proliferation [35]. The Orc1 or Orc2 deletion mutant cells have reduced Mcm2-7 helicase loading and their proliferation is critically dependent on Cdc6, an ATPase that is essential to form the ORC ring structure that encircles the duplex DNA. It remains unclear whether the ORC ring lacking Orc1 or Orc2 can still partner with Cdc6 to load enough Cdt1-Mcm2-7 complex for DNA replication or whether cells have ORC-independent and Cdc6-dependent alternative mechanisms to load the Cdt1-Mcm2-7 complex on DNA replication origins for replication licensing.

## 3. The Activity of Cdt1 Plays a Key Role in DNA Replication Licensing or Relicensing in Metazoans

Many studies reveal that Cdt1 acts as the loader of the Mcm2-7 replication helicase complex for pre-RC assembly to license DNA replication origins [3,36,37]. Among the replication initiation proteins involved in the pre-RC assembly, regulation of Cdt1 activity appears to be most critical for DNA replication licensing that ensures DNA replication occurs once and only once in a single cell cycle [1]. The Cdt1 activity is inhibited by binding to a specific inhibitory protein, Geminin [38], a protein initially identified as a protein substrate targeted for degradation by the Anaphase-promoting complex (APC/C) [39], a ubiquitin E3 ligase that targets mitotic proteins for ubiquitin-dependent degradation during the exit of mitosis and early G1. Replication licensing can occur in late M phase and early G1 when the protein level of Geminin is kept low by APC/C. During S phase to mitosis, the high level of Geminin protein inhibits Cdt1 replication licensing [40]. In metazoans, removal of Geminin is sufficient to induce the activation of Cdt1, promoting re-licensing and re-replication of genomes [41,42]. Crystal structure analysis revealed that the Cdt1-Geminin complex forms a hetero-hexamer, with the coiled-coil domain of Geminin interacting with Cdt1 so that the Cdt1 residues important for replication licensing are buried in the Cdt1-Geminin hexamer, preventing Cdt1 interaction with the Mcm2-7 helicase [43]. However, the Cdt1-Geminin complex appears to be dynamically regulated during DNA replication licensing. In Xenopus replication egg extract, depletion of Cdt1 inactivates the replication licensing activity. Notably, the addition of recombinant Cdt1-Geminin protein complex can activate replication licensing by recruiting the Cdt1-Geminin complex to chromatin to facilitate the interaction with the Mcm2-7 helicase even in the excessive amount of Geminin, whereas the addition of the recombinant Cdt1 protein alone without Geminin induces a checkpoint response that significantly slows down DNA replication [44]. These observations suggest that the Cdt1- Geminin complex may exist in the “permissive” or “inhibitory” conformations to regulate DNA replication licensing, as a recent crystal structure analysis revealed [45]. Further studies are required to understand the dynamic nature of the Cdt1-Geminin complex during DNA replication licensing and in S phase.

In addition to the removal of Geminin, artificial over-expression of Cdt1 protein is also sufficient to induce re-replication of the genome in mammalian cells [42]. Similarly, the addition of recombinant Cdt1 in Xenopus DNA replication egg extracts leads to the re-replication of DNA even in the G2 nuclei, which normally are refractory to DNA replication [46]. These induced DNA re-replications are usually accompanied with DNA damage-induced cell cycle checkpoint responses [41,42,47]. The critical replication licensing function of Cdt1 is also demonstrated in a mouse spontaneous 129 strain, in which a six-amino acid deletion in the PEST domain (amino acid residues 69–113) of the *Cdt1* gene causes the increased re-replication and transformation activities [48]. The expression of Cdt1 is frequently elevated in aggressive human hepatocellular carcinomas and in various breast and gastric carcinomas, suggesting aberrantly elevated levels of Cdt1 may contribute to these malignancies [40,49,50]. Conversely, hypomorphic mutations of the *Cdt1* encoding gene are associated with the Meier–Gorlin syndrome (MGS), a rare autosomal recessive primordial dwarfism disorder with microtia, patellar aplasia/hypoplasia, microcephaly, and short stature, similar to the mutational phenotypes of the *Orc1, Orc4, Orc6,* and *Cdc6* encoded proteins for the formation of the pre-RC complex for DNA replication [51,52,53].

Why is Cdt1 so unique among replication initiation proteins to promote DNA re-replication when it is over-expressed or aberrantly activated after the removal of Geminin? Recent studies revealed that there is a vast excess of Mcm2-7 helicase proteins relative to the ORC proteins assembled onto chromatin in G1 during the formation of pre-RC, although the loading of the Mcm2-7 helicase to the replication origins are dependent on the prior binding of the ORC-Cdc6 complex [54,55,56,57]. However, only a fraction of these Mcm2-7 proteins form the productive CMG helicase complexes for DNA replication unwinding at the onset of S phase [17]. Studies show that while the Cdt1-Mcm2-7 heptameric complex is loaded onto DNA cooperatively to form a double hexamer, Cdt1 interacts with Mcm6 and Mcm2 to stabilize the Mcm2-7 double hexamer helicase complex [7,26]. Since Cdt1 is required throughout the S phase, Cdt1 may act as a limiting step that stabilizes a fraction of large excessive Mcm2-7 proteins to load onto DNA for DNA replication.

Cdt1 also appears to modulate chromatin structure for replication licensing. In human cells, Hbo1 is a H4-specific histone acetylase (HAT) that interacts directly with Cdt1 for DNA replication licensing [58]. Hbo1 specifically interacts with replication origins in the G1 phase of the cell cycle and this association is dependent on the direct binding of Hbo1 to Cdt1 to enhance Cdt1-dependent DNA re-replication. Histone H4 acetylation at replication origins is cell cycle regulated, with maximal activity at the G1/S transition. The HAT-defective mutant of Hbo1 can bind to replication origins but the mutant is unable to load the Mcm2-7 helicase [9,58,59]. The requirement of Hbo1-mediated chromatin architecture for Mcm2-7 loading may be related to another recent finding that Grwd1, a glutamate-rich WD40 repeat protein that regulates chromatin openness by binding to histones, specifically interacts with several replication origins in G1 phase [60,61]. Grwd1 also specifically binds to Cdt1 to facilitate Mcm2-7 loading onto DNA. Genome-wide chromatin association analyses revealed that Grwd1 significantly colocalizes with Cdc6 in the genome [60,61]. Thus, Cdt1 may coordinate several important processes, including chromatin structural dynamics, to promote the loading of the double Mcm2-7 replication helicase hexamer complex onto chromatin DNA template for replication licensing in the cell cycle.

## 4. Regulation of Cdt1 Proteolysis by the CRL1 Ubiquitin Ligase Complex

The protein level of Cdt1 is dynamically regulated in the cell cycle [62]. It is high in G1 phase and low in S phase, although its mRNA level stays relatively constant in the cell cycle [62]. In the presence of 26S proteasome inhibitors, Cdt1 protein accumulates in S phase, suggesting that proteasome and ubiquitin-dependent proteolysis regulate the downregulation of Cdt1 protein levels in S phase [62]. Cdt1 was found to interact with the F-box protein, Skp2, a substrate-specific subunit of the cullin RING-based ubiquitin E3 ligase CRL1^Skp2^ (also called SCF^Skp2^) complex, and this interaction promotes the degradation of Cdt1 protein by CRL1^Skp2^ in the cell cycle [62,63,64]. The degradation of CRL1^SKP2^ substrates is usually phosphorylated and these site-specific phosphorylations are recognized by Skp2 [65,66]. Indeed, subsequent studies revealed that Cdt1 is phosphorylated by Cdk2 at threonine 29 (T29) and the phosphorylated T29 specifically binds to Skp2 to target Cdt1 for degradation by the CRL1^Spk2^ ubiquitin E3 ligase complex in the cell cycle. Although the T29-phosphorylated Cdt1 is degraded in S phase, several observations suggest that CRL1^Skp2^ is likely not to be the primary ubiquitin E3 ligase that critically regulates Cdt1 protein levels in S phase. First, the Cdt1 mutation that converts T29 to alanine (T29A), which cannot be phosphorylated by Cdk2 and fails to bind to Skp2, is still degraded normally in S phase [67]. There is no significant accumulation of the T29 mutant protein, as compared to the wild type Cdt1 protein, in S phase. Furthermore, the protein level of Cdt1 is not altered in fibroblasts lacking Skp2 (Skp2-/-) [64]. These findings indicate that CRL1^Skp2^ may contribute to the degradation of Cdt1 in S phase but it is not critical for the control of Cdt1 levels in S phase. Additional ubiquitin E3 ligase mechanism(s) exist that regulates Cdt1 protein degradation in the S phase of mammalian cells.

## 5. The CRL4 Ubiquitin Ligase Serves as the Primary E3 Ligase for Cdt1 Degradation in S Phase

Cullin 1 belongs to a large cullin protein family that also includes Cullin 2, 3, 4A and 4B, 5, and 7, all of which act as the critical scaffold proteins as the cullin RING-based ubiquitin E3 ligases (CRLs) [68]. Two original findings revealed that cullin 4 (CRL4) ubiquitin E3 ligase serves as the primary ubiquitin E3 ligase for the degradation of Cdt1 protein in S phase of the cell cycle. In Caenorhabditis elegans, loss of the single Cul4 orthologue leads to massive DNA re-replication, resulting in cells containing up to 100C DNA content [6]. While highly conserved Cdt1 orthologue protein is present in G1 cells, it disappears in S phase cells. Loss of Cul4 stabilizes Cdt1 protein in S phase. Removal of one genomic copy of Cdt1 gene suppresses the DNA re-replication phenotype caused by the loss of Cul4 [6], suggesting that Cul4-based ubiquitin E3 ligase may serve as the primary ubiquitin E3 ligase for Cdt1 degradation in S phase. Independently, in both human cancer cells and Drosophila S2 cells, UV- or gamma-irradiation was found to induce the rapid degradation of Cdt1 protein regardless of cell cycle [69], but not its associated inhibitory protein, Geminin. Screening of the cullin family proteins revealed that this DNA damage-induced Cdt1 degradation is prevented if Drosophila CUL4 is eliminated in response to DNA damage, while the loss of any other cullin E3 ligases does not suppress the degradation of Cdt1 after irradiation [69]. In human cells, loss of both Cul4A and Cul4B, two paralogues of Cul4, prevents Cdt1 degradation in response to UV or gamma irradiation [69]. These studies establish that CRL4 ubiquitin E3 ligase regulates Cdt1 degradation in S phase and in response to DNA damage. Stabilization of Cdt1 protein after the loss of Cul4 in S phase promotes DNA re-replication. Notably, the DNA damage induced Cdt1 degradation occurs even when the ATM/ATR-mediated DNA damage checkpoint control is inhibited, suggesting the presence of an alternative DNA damage mechanism for the degradation of Cdt1 [69].

## 6. Cdt1 Interacts with Proliferation Cell Nuclear Antigen (PCNA) for Degradation

If Cdt1 is degraded in S phase cells or in response to DNA damage, what might be the common mechanism by which Cdt1 is targeted for degradation under these different conditions? Cdt1 was found to contain a highly conserved PCNA-interacting peptide box (PIP) at its extreme amino terminus (Figure 1A) [5,70,71]. This PIP box indeed mediates the direct interaction between Cdt1 and PCNA (proliferation cell nuclear antigen) [5]. In the in vitro Xenopus activated egg DNA replication system, the CRL4-dependent Cdt1 degradation is recapitulated by the in vitro DNA replication or DNA damage-induced repair replication process. Antibody-based depletion of PCNA from the extract is sufficient to inactivate Cdt1 degradation induced by damaged DNA in the extract [5]. In mammalian cells, mutation or deletion of this PIP box in Cdt1 prevents its degradation in response to DNA damage [70,71]. Since PCNA encircles DNA as a trimeric ring that serves as a clamp to facilitate the processivity of DNA polymerases during DNA replication and damage-induced repair DNA synthesis [1], these studies provide a common mechanism by which DNA replication and DNA damage trigger the degradation of Cdt1 protein through its direct interaction with PCNA [1].

## 7. Cdt2 Serves as the Substrate-Specific Receptor for CRL4-Mediated Degradation of Cdt1

The CRL4 complex, like other cullin family ubiquitin E3 ligases such as CRL1^Skp2^, employs a modular mechanism using one of many substrate-specific receptor proteins to interact with specific protein substrates for ubiquitin-dependent degradation [68]. While the carboxy terminal regions of Cul4A or its paralogue Cul4B binds to the RING protein Rbx1 or Rbx2 (also called Roc1 or Roc2) for the binding of ubiquitin conjugating E2 enzyme for substrate ubiquitination, the amino terminal region of Cul4A or Cul4B binds to the adaptor protein, DNA damage binding protein 1 (Ddb1), to form the CRL4 core complex [72]. Loss of Ddb1, similar to Cul4A and Cul4B, also blocks the DNA damage-induced Cdt1 proteolysis [73]. It was found that a set of WD40 repeat domain proteins, DCAFs (also called CDWs/DWDs), serve as the substrate-specific receptor subunits that interact with DDB1 to form various CRL4^Dcaf^ ubiquitin E3 ligase complex to target specific substrates for poly-ubiquitination and subsequent degradation [74,75,76,77,78,79]. Cdt2 (also called L2dtl/Dtl/Dcaf2) was isolated as a WD40 domain-containing protein associated with the purified CRL4 core proteins Cul4B or Ddb1 in human cancer cells [74,75,78]. Loss of Cdt2, but not other Dcaf proteins, specifically blocked the degradation of Cdt1 induced by DNA replication or DNA damage, and induced re-replication associated with an increased nuclear volume, a cell population with >4N DNA content, and altered DNA sedimentation [74,75]. Cdt2 is evolutionarily conserved from yeast, C. elegans, Drosophila, and humans to regulate Cdt1 degradation by directly interacting with Cdt1 [1,74]. These studies indicate that Cdt2 is the substrate-specific subunit of the CRL4^Cdt2^ ubiquitin E3 ligase complex for Cdt1 degradation [1,79]. In addition to preventing Cdt1-mediated DNA re-replication, CRL4^Cdt2^ ubiquitin E3 ligase complex also controls Drosophila endocycle DNA replication, a variant cell cycle with only S and G2 but not mitosis in certain Drosophila cells [80].

Since Cdt1 contains a PIP box, which serves as a degron motif for its degradation by PCNA- and CRL4^Cdt2^-dependent proteolysis, subsequent studies reveal that many PIP box-containing proteins, including CDK inhibitor p21 (CDN1A or CIP1/WAF1) or its Xenopus orthologue Xic1 [81,82,83], Set8 (also called KMT5A, PR-SET7/9, SETD8) histone methyltransferase [84,85,86,87], thymine DNA glycosylase TDG [88,89], Drosophila E2F1 [90], the p12 subunit of DNA polymerase δ [91], Xeroderma pigmentosa group G (XPG) [92], and F-box DNA helicase 1 (FBH1) [93], are the proteolytic substrates of CRL4^Cdt2^ in response to DNA damage or in S phase for the PCNA-dependent proteolysis (Figure 1A). The C. elegans DNA polymerase η contains a conserved PIP box and is also regulated by CRL4^Cdt2^ [94]. While the degradation of Cdt1 prevents the re-licensing of replication origins to ensure DNA replication occurs only once in the cell cycle, the significance of other CRL4^Cdt2^-mediated substrate degradation has emerged. For example, the PCNA-dependent degradation of Cdk inhibitor p21 by CRL4^Cdt2^ is important because the presence of p21 impairs DNA damage-induced replication, which is associated with a deficiency for recruitment of DNA polymerases from the Y family members involved in translesion DNA synthesis, resulted in the accumulation of DNA damage markers and genome instability [95]. Set8 is a histone methyltransferase that methylates lysine 20 in histone H4 (H4K20), which further regulates chromatin compaction and replication licensing of replication origins [87,96]. Recent studies showed that a chromatin compaction threshold exists in cells exiting mitosis to ensure genome integrity by limiting DNA licensing in G1 phase [97]. Upon mitotic exit, Set8 regulates chromatin relaxation. Loss of Set8 leads to substantial genome-wide chromatin decompaction and allows excessive loading of the ORC complex to replication origins in daughter cells. ORC overloading to replication origins stimulates aberrant recruitment of Mcm2-7 helicase complex that promotes single-strand DNA formation and DNA damage [97]. Thus, the CRL4^Cdt2^ ubiquitin E3 ligase regulates multiple processes important for replication licensing, DNA replication, and cell cycle progression by targeting PIP-containing protein substrates, including Cdt1, p21, and Set8 for ubiquitin-dependent degradation.

However, many other PIP box-containing proteins, such as DNA ligases, Fen1 (Flap endonuclease 1), and other DNA polymerases, are not substrates for CRL4^Cdt2^ [1]. Comparison between various PIP boxes in CRL4^Cdt2^ substrates and non-substrate proteins suggest that the presence of positively charged amino acid residues near the PIP box may help define the PIP box degron motif in CRL4^Cdt2^ substrates [1,98,99]. It was proposed that Cdt1 and other CRL4^Cdt2^ substrates are recruited onto chromatin through their interaction with the DNA-bound trimeric PCNA clamp on replicating DNA in S phase or during DNA damage repair to be targeted for degradation by the chromatin bound CRL4^Cdt2^ [5], although how CRL4^Cdt2^ is recruited onto the replicating chromatin to target Cdt1 and other PIP box-containing substrates for degradation remains unclear at the time (see below).

## 8. Cdt2 Also Contains a PIP Box-Like Motif That Mediates the Direct Interaction with PCNA to Target Cdt1 Degradation in S Phase or in Response to DNA Damage

Although the CRL4^Cdt2^ E3 ligase complex regulates the degradation of the PIP box-containing proteins such as Cdt1, p21, or Set8, several lines of emerging evidence indicate that this ubiquitin E3 ligase is also recruited onto the replicating chromatin by PCNA. First, during the affinity purification of the Cdt2 protein complex from human cancer cells, a small number of PCNA protein fragments were found in the Cdt2 complex [74]. Conversely, affinity-purified PCNA complex also reveals the presence of Cdt2-derived peptides among other known PCNA-associated proteins such as RFC1 [100]. These reciprocal studies indicate that Cdt2 may interact with PCNA. Subsequent biochemical studies reveal that recombinant Cdt2 and PCNA proteins can directly interact with each other via a highly conserved PIP box-like motif at the carboxy-terminus of Cdt2 (Figure 1B). Mutation of this PIP box-like motif in Cdt2 abolishes the interaction between CDT2 and PCNA both in vitro and in vivo. While the introduction of wildtype Cdt2 can rescue the Cdt1 degradation defects in Cdt2 knockdown cells, the PIP-box mutant of Cdt2 cannot substitute the normal Cdt2 function for Cdt1 degradation in S phase or in response to DNA damage [100]. These studies are consistent with a previous report that the carboxy half of Xenopus Cdt2 homologue is required for Xenopus Cdk inhibitor Xic1 protein degradation in Xenopus egg extract and this requirement is associated with the ability of this region to interact with PCNA [83]. Independently, the C-terminal region of Cdt2 is found to help recruit Cdt2 to the DNA damage sites in mammalian cells and the same PIP box-like region in Cdt2 is found to mediate the high affinity interaction between Cdt2 and PCNA [101]. These studies suggest a model by which both the PIP box-containing substrate proteins, such as Cdt1, and the CRL4^Cdt2^ ubiquitin E3 ligase are recruited to the replicating DNA by directly interacting with different monomeric subunits of the trimeric PCNA clamp during DNA replication. By binding to the same trimeric PCNA clamp, the proximity of Cdt2 and its substrates leads the substrate degradation during DNA replication in S phase or during DNA damage-induced DNA repair synthesis (Figure 2) [100].

## 9. Regulation of the CRL4^Cdt2^ Ubiquitin E3 Ligase

The protein levels of Cdt2 are dynamically regulated in the cell cycle. For example, Cdt2 is found to be phosphorylated by Cdks on threonine 464 (T464) and this phosphorylation event prevents the ubiquitin-dependent proteolysis of Cdt2. A CRL1-based ubiquitin E3 ligase, CRL1^FBXO11^, was identified, which targets the unphosphorylated Cdt2 for degradation during the control of the cell cycle exit [102,103]. However, it remains to be investigated whether additional mechanisms exist to regulate CRL4^Cdt2^ ubiquitin E3 ligase for the degradation of Cdt1 and many other substrates in the cell cycle.

## 10. Conclusions

The CRL4^Cdt2^-mediated proteolysis of replication licensing protein Cdt1 after replication initiation has emerged as the key mechanism that prevents DNA re-replication and genome instability in a single cell cycle. Recent studies indicate that the trimeric PCNA clamp that encircles replicating DNA strands recruits both Cdt1 and Cdt2-associated CRL4 ubiquitin E3 ligase complex to target Cdt1 for ubiquitin-dependent degradation in the S phase or during DNA damage-induced repair synthesis. Further studies are required to determine how the PCNA-bound Cdt2 selectively interacts with Cdt1 and other substrates of CRL4^Cdt2^ for PCNA-dependent proteolysis.

## Figures and Tables

**Figure 1 ijms-22-05195-f001:**
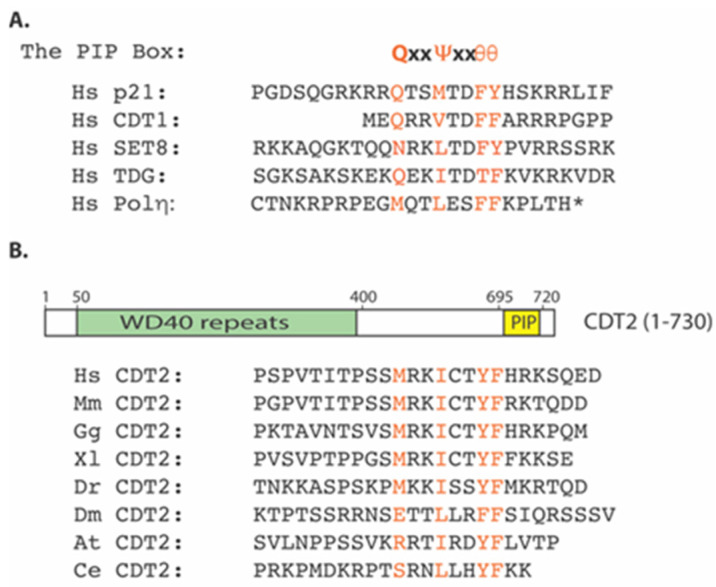
Both Cdt2 and its identified protein substrates contain the conserved PIP box motifs. (**A**) The consensus PIP box motifs in various CRL4^Cdt2^ substrates. (**B**) The presence of a conserved PIP box-like motif in the Cdt2 proteins from different species.

**Figure 2 ijms-22-05195-f002:**
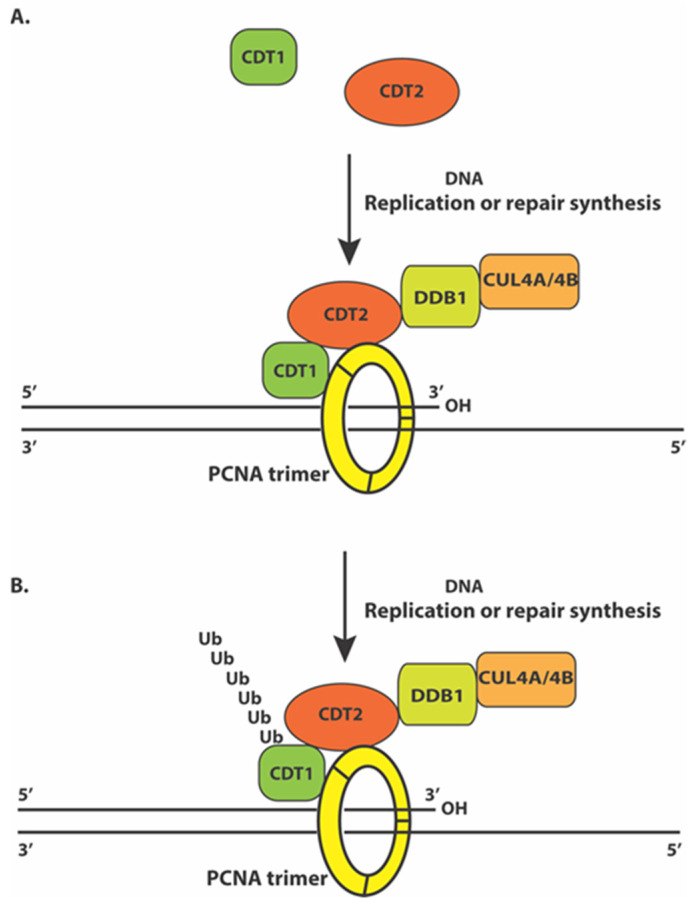
A model for DNA synthesis and PCNA-dependent Cdt1 degradation by CRL4^CDT2^. (**A**) During DNA replication or DNA damage-induced repair synthesis, Cdt1 and Cdt2 are recruited onto different subunits of the trimeric PCNA clamp encircling the replicating DNA strands. (**B**) The binding of Cdt1 and Cdt2 to the same trimeric PCNA clamp during DNA synthesis promotes the specific interaction between Cdt1 and CRL4^Cdt2^ for the ubiquitin-dependent proteolysis of Cdt1 to prevent DNA re-replication and genome instability.

## Data Availability

Not applicable.

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
