# Peer review of "Regulation of DNA Replication Licensing and Re-Replication by Cdt1"

_ijms, 2021, doi:10.3390/ijms22105195_

Round 1
Reviewer 1 Report
This review on replication goes into detail of the role on CDT1 and it's role in replication.
The authors appear to use the appropiate citations through-out the course of the manuscript.
The figures includes are required for the manuscript.
There are no major concerns with this manuscript.
Author Response
I would like to thank the reviewer for positive comments on the manuscript. I have corrected minor spelling mistakes such as “OCR” in the revised manuscript as suggested by the reviewer.
Reviewer 2 Report
The “ Regulation of DNA Replication Licensing and Re-replication” manuscript is well written and very clear. It addresses a very important process for maintaining genome integrity. It is a very characterized topic, and this is also evidenced by the fact that most of the bibliography is somewhat historical, but it picks up on important recently identified regulatory mechanisms that leave open other regulatory processes possible.
Probably a new title more focused on regulation of CDT1 could be more appropriate.
There are in general several problems with the nomenclature, especially in the first part when referring to information about S. cerevisiae (which should be written in italic like the other species mentioned), this sometimes creates confusion and you don't understand if you are talking about the mechanism of yeast or mammalian cells. In particular the genes of S.cerevisiae should be written in capital italic if WT and in small italic if mutated. When talking about proteins they are normally written with the first letter capitalized and optionally you can indicate the p at the end of the name. See SGD site for more details.
pag2 (line 47) the orthologue of CDT1 according to SGD should be TAH11, no TAH2. Devault A, et al. (2002) Identification of Tah11/Sid2 as the ortholog of the replication licensing factor Cdt1 in Saccharomyces cerevisiae. Curr Biol 12(8):689-94
Line 96: CDT-MCM2-7 complex instead CDT1-MCM2-7 complex, and other typos such as OCR instead ORC in the text.
page 3 geminin-CDT1 is sometimes written as CDT1-geminin and with capital G
It is not very clear why this example is included (page 5 lines 213-215):
“such as the phosphorylation of threonine 187 (T187) in the CDK inhibitor p27Kip1 (CDKN1B) by cyclin/CDKs, triggering their proteolysis in the cell cycle (65, 66).”
Page 5 line 252 “suggesting the presence of a novel DNA damage mechanism for the degradation of CDT1 (69).”
The citation comes from a 2003 work, perhaps I would say alternative, rather than novel.
Line 316 page 7 citation 95: is cited as recently emerged data, but it is from 2013
Line 337. This conclusion is not very clear since a model for this point is proposed in the next paragraph (100):
“However, the important question how CRL4CDT2 is recruited onto replicating chromatin to target CDT1 and other PIP box-containing substrates for degradation remains unclear.”
Author Response
I would like to thank the reviewer for careful reading of the manuscript and excellent suggestion and comments. Here are my responses to reviewer’s comments:
Probably a new title more focused on regulation of CDT1 could be more appropriate.
Response: Many thanks for reviewer’s helpful comments. I agree and have changed the title to “Regulation of DNA Replication Licensing and Re-replication by CDT1”
There are in general several problems with the nomenclature, especially in the first part when referring to information about S. cerevisiae (which should be written in italic like the other species mentioned), this sometimes creates confusion and you don't understand if you are talking about the mechanism of yeast or mammalian cells. In particular the genes of S.cerevisiae should be written in capital italic if WT and in small italic if mutated. When talking about proteins they are normally written with the first letter capitalized and optionally you can indicate the p at the end of the name. See SGD site for more details.
Response: Many thanks for reviewer’s helpful corrections. I have changed S.cerevisiae and Schizosacharomyces pombe to italic S.cerevisiae and Schizosacharomyces pombe and yeast proteins using the first letter capitalized such as CDT1 to Cdt1 throughout the revised manuscript.
pag2 (line 47) the orthologue of CDT1 according to SGD should be TAH11, no TAH2. Devault A, et al. (2002) Identification of Tah11/Sid2 as the ortholog of the replication licensing factor Cdt1 in Saccharomyces cerevisiae. Curr Biol 12(8):689-94
Response: Many thanks for reviewer’s helpful correction again. I have changed Tah2/Sid2 to Tah11/Sid2 in the revision.
Line 96: CDT-MCM2-7 complex instead CDT1-MCM2-7 complex, and other typos such as OCR instead ORC in the text.
Response: Many thanks for reviewer’s helpful correction again. I have corrected the CDT-MCM2-7 complex mistake to CDT1-MCM2-7 complex. I also corrected OCR to Orc in the text.
page 3 geminin-CDT1 is sometimes written as CDT1-geminin and with capital G
Response: Many thanks for reviewer’s helpful comments. I have changed geminin-CDT1 to CDT1-Geminin and replaced all geminin by Geminin.
It is not very clear why this example is included (page 5 lines 213-215):
“such as the phosphorylation of threonine 187 (T187) in the CDK inhibitor p27Kip1 (CDKN1B) by cyclin/CDKs, triggering their proteolysis in the cell cycle (65, 66).”
Response: Many thanks for reviewer’s helpful comments. I agree that this sentence might create some confusion. I have deleted this example and the revised sentence now is: “The degradation of CRL1SKP2 substrates is usually phosphorylated and these site-specific phosphorylations are recognized by SKP2 (65, 66).”
Page 5 line 252 “suggesting the presence of a novel DNA damage mechanism for the degradation of CDT1 (69).”
The citation comes from a 2003 work, perhaps I would say alternative, rather than novel.
Response: Many thanks for reviewer’s helpful comments again. I have changed “novel” to “alternative”:
“suggesting the presence of an alternative DNA damage mechanism for the degradation of CDT1 (69)” in the revision.
Line 316 page 7 citation 95: is cited as recently emerged data, but it is from 2013
Response: Many thanks for reviewer’s helpful comments again. I have removed “recently” so the sentence now is “the significance of other CRL4CDT2-mediated substrate degradation has emerged”
Line 337. This conclusion is not very clear since a model for this point is proposed in the next paragraph (100):
“However, the important question how CRL4CDT2 is recruited onto replicating chromatin to target CDT1 and other PIP box-containing substrates for degradation remains unclear.”
Response: Many thanks for reviewer’s helpful comments again. I have rephased the sentence to “although how CRL4CDT2 is recruited onto replicating chromatin to target CDT1 and other PIP box-containing substrates for degradation remains unclear at the time (see below).”
This manuscript is a resubmission of an earlier submission. The following is a list of the peer review reports and author responses from that submission.